**Data availability statement:** All source code necessary to reproduce all simulations are provided are available in the Zenodo repository: https://zenodo.org/records/15866455.

# Impact of exposure frequency on disease burden of the common cold – A mathematical modeling perspective

**Sebastian Gerdes**[☉]*, **Michael Rank**[☉], **Ingmar Glauche, Ingo Roeder**[iD]

Institute for Medical Informatics and Biometry, Faculty of Medicine Carl Gustav Carus, TUD Dresden University of Technology, Fetscherstraße, Dresden, Germany

☉ These authors contributed equally to this work.

* sebastian.gerdes@tu-dresden.de

## Abstract

The common cold is a frequent disease in humans and can be caused by a multitude of different viruses. Despite its typically mild nature, the high prevalence of the common cold causes significant human suffering and economic costs. Oftentimes, strategies to reduce contacts are used in order to prevent infection. To better understand the dynamics of this ubiquitous ailment, we develop two novel mathematical models: the common cold ordinary differential equation (CC-ODE) model at the population level, and the common cold individual-based (CC-IB) model at the individual level. Our study aims to investigate whether the frequency of population / individual exposure to an exemplary common cold pathogen influences the average disease burden associated with such a virus. Results derived for this situation can also be applied to other common cold pathogens.

On the one hand, the CC-ODE model captures the dynamics of the common cold within a population, considering factors such as infectivity and contact rates, as well as development of specific immunity in the population. On the other hand, the CC-IB model provides a granular perspective by simulating individual-level interactions and infection dynamics, incorporating heterogeneity in contact rates. In both modeling approaches, we show that under specific parameter configurations (i.e., characteristics of the virus and the population), increased exposure can result in a lower average disease burden. While increasing contact rates may be ethically justifiable for low-mortality common cold pathogens, we explicitly do not advocate for such measures in severe illnesses. The mathematical approaches we introduce are simple yet powerful and can be taken as a starting point for the investigation of specific common cold pathogens and scenarios.

## 1 Introduction

### 1.1 Background

The common cold is a frequent disease in humans and is generally caused by viral infection of the upper respiratory tract [1]. Although mild in most cases, it poses a significant disease burden on individuals and societies, both in terms of human suffering and economic loss.

**Funding:** The author(s) received no specific funding for this work.

**Competing interests:** The authors have declared that no competing interests exist.

The common cold is the most frequent illness in the US with approximately 25 million documented cases per year [2]. It is estimated that in the US alone, the economic cost of the common cold is approximately $25 billion per year [3]. Despite the large number of cases and the great associated disease and economic burden the common cold is currently not a prioritized research topic. A thorough and precise understanding of the disease dynamics both on a societal and an individual level are lacking today, but might pave the way to better prevention and treatment strategies.

The most frequent causative viral agents are rhino viruses (approx. 30% to 50%), corona viruses (approx. 10% to 15%, not including SARS-CoV-1, SARS-CoV-2 and MERS), influenza viruses (approx. 5% to 15%), respiratory syncytial viruses (approx. 5%), parainfluenza viruses (approx. 5%), adeno viruses (less than 5%), entero viruses (less than 5%), and further unknown viruses (approx. 20% to 30%) [4]. Of these viruses, different strains are circulating and they are constantly subjected to genetic shift and drift.

The immune system is generally capable of clearing a common cold without additional treatment. In healthy individuals the symptoms are often relatively mild (including sneezing, stuffy nose, runny nose, sore throat, coughing, post-nasal drip, watery eyes, fever). Most often, the intensity of symptoms peaks around day 3 or 4 and around day 7 recovery begins [4]. The median duration of symptoms of a common cold has been estimated to be approximately 11 days [5].

In fighting a common cold, different components of the immune system are involved [6]. Generally, the infectious agent enters via the mucous membranes. Here, both specific and unspecific components of the immune system can often already eliminate the infectious agent. In this case, the exposure may result in an asymptomatic course, however, possibly involving training of the immune system. If the infectious agent settles and proliferates, typical symptoms of a common cold may develop. Over time, more powerful components of the specific immune system come into play. In particular, the specific immune system continuously improves its ability to recognize the pathogen and efficiently eliminates it.

After the infection has subsided, the immune system usually retains the ability to recognize the respective pathogen for a while. Hence, after immediate reexposure, it is unlikely that another symptomatic infection occurs. However, with time the newly acquired specific immunity generally deteriorates and may even revert to the baseline level. Furthermore, it is important to note that there is significant cross-reactivity between different virus strains, e.g. in the case of rhinoviruses [7], and possibly even between different viruses. Therefore, a symptomatic infection may be alleviated or prevented after exposure to a virus, if an infection with a similar virus or virus strain has occurred previously (Fig 1).

During the COVID-19 pandemic, many regions were subject to long-lasting, extensive contact restrictions, particularly in the winters of 2020/2021 and 2021/2022. Consequently, the incidence of common colds has dropped during this time. The reduced training of immunity to common colds has presumably been linked to a general decline in the population's immunity to cold viruses [8,9]. In the winter 2022/2023, an increased number of sick days due to common colds was observed in the communities where the contact rates had been lower in the preceding years (e. g. in Germany [10]).

The degree of persistence of specific immunity seems to differ considerably between different common cold pathogens [11]. Infections with rhinoviruses and adenoviruses seem to generally result in long-term, protective immunity against the specific virus serotype, however, not against other serotypes of the virus. Infections with coronaviruses, parainfluenzaviruses, RS-viruses and multiple other common-cold-viruses seem to usually result in short-term immunity, that declines over time.

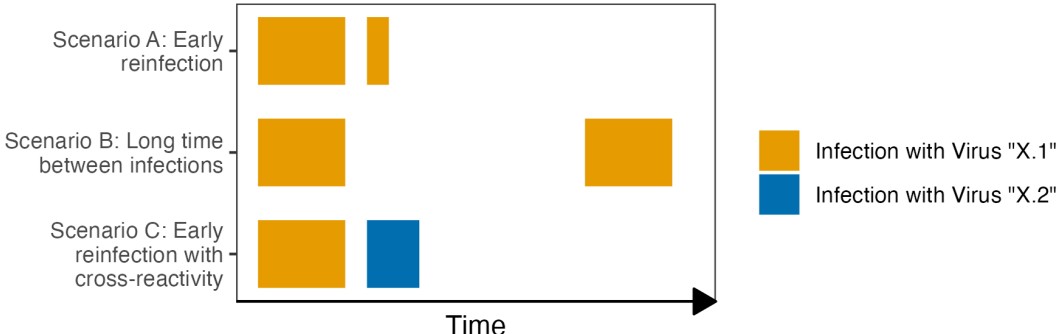

**Fig 1. Schematic representation of different hypothetical scenarios for immunity upon reinfection with an exemplary virus denoted by "X" on a time scale without units.** In scenario A, upon immediate reinfection with strain "X.1" of the exemplary virus "X", symptoms are likely to be mild and the infection is usually short or the infection might even be asymptomatic. In scenario B the time between infections is long, and immunity will likely be lost for many viruses and duration of the infection would be long again and might be associated with more severe symptoms. Scenario C considers reinfection with an exemplary strain "X.2" of virus "X". In this case there might be some cross-reactivity, which alleviates symptoms in case of infection with the related virus strain "X.2".

There is a large corpus of mathematical models describing the dynamics of infectious diseases in a population. Among these models, the classical Susceptible-Infected-Recovered (SIR) epidemic model introduced by [12] holds a prominent place. It is an ordinary differential equation (ODE) model, in which a susceptible fraction of the population (compartment **S**) can get infected with an infectious disease and hence transition to compartment **I**. Finally, individuals in compartment **I** recover and hence pass into compartment **R**, in which they are immune to the infectious disease. There are numerous variants of this model. One variant that is relevant in the context of this paper is the SIRS model, in which immunity in compartment **R** is lost with a certain rate and individuals hence transition from compartment **R** back into compartment **S**. Another variant of the SIR model is the SIS model, described e. g. by [13,14], in which no immunity is acquired (no compartment **R**) and individuals pass directly into compartment **S** when recovering.

In the classical SIRS model, the loss of immunity occurs instantaneously after transitioning from **R** to **S** and is hence a binary (immune / not-immune) rather than a gradual property. Assuming a constant transition rate, the resulting duration of immunity is exponentially distributed in a probabilistic setting. Heffernnan and Keeling [15] have examined modifications to this model, examining different waning schemes where immunity is lost gradually rather than abruptly. Specifically they studied stochastic SIR-like models. They conclude, that faster waning of immunity leads to a higher disease prevalence in the population. Khalifi and Britton [16] studied further variants of immunity waning within deterministic ODE, in particular also SIRS-like, models. In a subsequent study, Khalifi and Britton [17] explored optimizing vaccine schemes in the context of immunity waning in SIR-like models and provided recommendations for effectively achieving herd immunity.

The common cold syndrome may be caused by a multitude of different cross-reacting pathogens, potentially resulting in virus-specific and for some viruses cross-virus immunity. These complex immunological interactions between different viruses have been studied using several mathematical SIR-type models (e. g. [18], [19], [20]).

Waning of immunity following SARS-CoV-2 infection or vaccination has been studied in detail, providing quantitative estimates for the temporal decline in immune protection [21]. Recent research has further examined how the frequency of reinfection depends on the rate

of immune waning, offering insights into long-term epidemiological dynamics [22]. However, these studies have primarily focused on incidence patterns without explicitly assessing the resulting overall disease burden.

A modified SEIR model incorporating explicit transmission dynamics, waning immunity, behavioral responses, and vaccination strategies has also been developed, demonstrating that long-term social confinement, reduced contact rates, and moderately effective vaccination programs can significantly reduce infection peaks [23]. Yet, also in this study, the net disease burden was not explicitly quantified.

In a pathogen-agnostic approach, it has been analyzed how immunity acquisition and waning of immunity impact on expected death rates [24]. Even there, the net disease burden of non-fatal courses has not been analyzed.

## 1.2 Objectives

The relation between disease burden per capita per time (referred to as *mean disease burden* from hereon) and the exposure frequency to particular viruses typically causative of common colds has not been explicitly studied. Exposure frequency may depend among other aspects on overall interpersonal contact patterns and contact reduction strategies of infected individuals. In a certain range, increased exposure likely increases the spread of viral disease and, thus, increases the mean disease burden for most viruses causing common colds. However, we want to investigate if this relation might reverse under certain circumstances, if a critical exposure frequency is exceeded due to increased training of the immune system. In other words, more contacts and hence more exposure to pathogen might strengthen the immune system so that for some pathogens the mean disease burden related to this virus might be reduced. Even though this seems plausible, there are neither reliable epidemiological data available demonstrating such an effect for the common cold, nor has this aspect been studied explicitly in a mathematical modeling framework to our knowledge.

Therefore, we aim to investigate this hypothesis by applying a mathematical modeling approach. Mathematical models express hypotheses in formal, quantitative terms and can be used to evaluate the implications of different hypotheses and design informative experiments. In the following, we present two novel, related mathematical models in order to address the aforementioned research question.

In our analysis, we consider an exemplary virus that can cause an upper respiratory infection. Although the common cold can be caused by a wide variety of viruses, the clinical manifestations—particularly the duration and characteristics of symptoms—often show substantial overlap. Our modeling approach is designed to capture these shared features, allowing us to describe the typical behavior of common cold infections within a generalized framework. However, we acknowledge that the underlying immunological responses can differ significantly across pathogens, particularly with respect to the development and duration of pathogen-specific immunity. To maintain conceptual clarity and analytical tractability, we deliberately neglect immunological interactions between different pathogens. This simplification allows us to focus on core mechanisms while preserving flexibility: the biological properties of the modeled pathogen, such as infectivity and recovery dynamics, can be adjusted within the framework to represent a range of plausible scenarios.

The proposed methods can be applied to any real common cold pathogen by choosing appropriate parameter configurations or estimating them from respective data. Our goal is to find a simple and universal mathematical framework that describes the progression of a typical common cold rather than the precise analysis of a specific pathogen.

Our work is based on the prototypical SIR model family, which allows for an easy connection and integration into the current scientific discourse. The first model that we call **CC-ODE model** (*common cold ordinary differential equation model*) is an ODE model based on the SIRS model. In contrast to the plain SIR model, the SIRS model allows to describe loss of immunity. A formal description of the CC-ODE-model can be found in Sect 2.1 and analysis results in Sect 3.1. The second model is an individual-based model derived from the SIS model and is referred to as **CC-IB model** (*common cold individual-based model*). Individual-based models allow to follow individuals and their properties [25]. In the CC-IB model, immunity is represented as an individual-specific state. Hence, the explicit modeling of the **R** compartment is not necessary and the SIS model is sufficient as a basis. The CC-IB model is described in Sect 2.2 and Sect 3.2.

The two chosen modeling approaches shall complement each other, the CC-ODE model being better suited for deriving analytical results in closed form and deriving population-centered results, while the CC-IB model inherently allows to represent heterogeneity among individuals, stochasticity and tracking of individual fates.

## 2 Methods

### 2.1 CC-ODE model

In this section, the **CC-ODE model** (*common cold ordinary differential equation model*) is presented and described. It is an ODE model based on the SIRS model, a classical model describing the dynamics of an infectious disease on a population level. In the SIRS model, individuals can switch between three different compartments. First, the susceptible individuals are in compartment **S**. They do not carry the disease and can potentially get infected. The number of individuals in this compartment at time $t$ is given by $S = S(t)$. Second, the infected individuals are in compartment **I** and can spread the disease. The number of individuals in this compartment at time $t$ is denoted by $I = I(t)$. Third, the recovered (and immune) individuals are in compartment **R**. They can neither acquire nor spread the disease and the number of individuals at time $t$ reads $R = R(t)$. The total number of individuals $N(t) = S + I + R$ is fixed (phenomena such as birth, death and migration are not included in the model). For reasons of simplicity, we assume $N = 1$ so that $S$, $I$ and $R$ can be interpreted as proportions of a population that is constant in size. The model equations read

$$\frac{dS}{dt} = -\beta IS + \delta R$$
$$\frac{dI}{dt} = \beta IS - \gamma I$$
$$\frac{dR}{dt} = \gamma I - \delta R$$

with initial conditions $S(0)$, $I(0)$ and $R(0)$. The model parameters are

- $\beta \geq 0$: infection rate (influenced by contact rate and infectivity of the pathogen),
- $\gamma \geq 0$: recovery rate,
- $\delta \geq 0$: immunity loss rate.

Note that by setting the immunity loss $\delta = 0$ we obtain the basic SIR model. This model describes the dynamics of a population exposed to one single virus without interaction to other pathogens or concurrent virus strains. An infection with this virus (strain) can lead to immunity that eventually subsides, which holds true for most common cold pathogens.

The basic reproduction number of this model is $\mathcal{R}_0 = \frac{\beta}{\gamma}$. The model implies a steady state solution in which the fraction of infected individuals persists at a constant level over time (endemic solution / equilibrium), which reads:

$$\begin{pmatrix} S^* \\ I^* \\ R^* \end{pmatrix} = \begin{pmatrix} \dfrac{\gamma}{\beta} \\ \dfrac{\delta(\beta - \gamma)}{\beta(\gamma + \delta)} \\ \dfrac{\gamma(\beta - \gamma)}{\beta(\gamma + \delta)} \end{pmatrix} = \begin{pmatrix} \dfrac{1}{\mathcal{R}_0} \\ \dfrac{\delta(\mathcal{R}_0 - 1)}{\mathcal{R}_0(\gamma + \delta)} \\ \dfrac{\gamma(\mathcal{R}_0 - 1)}{\mathcal{R}_0(\gamma + \delta)} \end{pmatrix}$$

To investigate if a higher contact rate can eventually lead to a reduced mean disease burden, we introduce some amendments of the equations and parameters leading to an ODE model, which we call *CC-ODE model*. We assume that an increase of the contact rate leads to shorter (or less intense) infections due to development of specific immunity. Hence, the higher the contact rate, the higher also the recovery rate. However, there is no evidence that the infectivity of the pathogen should directly affect the recovery rate. This is why we split the infection rate $\beta$ into two independent factors that describe the infection rate by a contact rate ($\beta_2$) and a measure of infectivity ($\beta_1$), which can be interpreted as the probability of infection upon exposure:

$$\beta = \beta_1 \cdot \beta_2.$$

In order to represent the possibility of development of specific immunity to the pathogen (= habituation effect), we introduce the parameter $\alpha$ representing the immunogenicity of the virus, giving rise to an additional term $\alpha\beta_2 I$ in the model equations:

$$\frac{dS}{dt} = -\beta_1\beta_2 IS + \delta R$$
$$\frac{dI}{dt} = \beta_1\beta_2 IS - (\gamma + \alpha\beta_2)I$$
$$\frac{dR}{dt} = (\gamma + \alpha\beta_2)I - \delta R$$

with the additional model parameters

- $\beta_1 \geq 0$: infectivity,
- $\beta_2 \geq 0$: contact rate,
- $\alpha \geq 0$: immunogenicity, i. e. a measure of the degree of development of specific immunity to the pathogen in the population upon exposure.

$\beta_1$ measures the infectivity of the pathogen. Large values of $\beta_1$ correspond to highly contagious diseases. The contact rate $\beta_2$ describes the number of contacts per individual in society, with high values indicating a very active population with many contacts. In this way, the higher both of these parameters, the higher the total infection rate. If the infectivity $\beta_1 = 0$ (non-contagious disease) or contact rate $\beta_2 = 0$ (absolute isolation), there is no spread of the disease. The parameter $\alpha$ describes the immunogenicity of the virus, i. e. the immunological habituation effect of the specific immune system against the virus in question. For $\alpha = 0$, there is no development of specific immunity at all and we obtain the classical SIRS model.

For immunogenicity $\alpha \to +\infty$ and contact rate $\beta_2 > 0$, development of specific immunity is so effective that infections are eliminated immediately.

In the current version of the model, immune system boosting depends solely on the contact rate ($\beta_2$), not on $\beta_1$ (the probability of infection per contact). This reflects the assumption that immune stimulation may occur even in the absence of full infection and significant viral replication—i.e., that exposure alone can be sufficient to trigger an immune response.

This model is applied to describe the dynamics of the common cold on a *population level*. We aim to investigate the disease dynamics in the CC-ODE model by introducing an infection into a small proportion of a population without prior immunity. Thus, the initial conditions read $S(0) = 0.99$, $I(0) = 0.01$ and $R(0) = 0$ for all our simulations of the CC-ODE model.

## 2.2 CC-IB model

The **CC-IB model** (*common cold individual-based model*) is a novel agent-based model derived from the SIS model. The number of individuals is denoted to $N$ ($N$ is an integer > 1 in contrast to the previous section where $N$ was fixed to 1). The dynamics of the SIS model are described by the following equations:

$$\frac{dS}{dt} = -\beta S \frac{I}{N} + \gamma I$$
$$\frac{dI}{dt} = \beta S \frac{I}{N} - \gamma I$$

As in the SIR model, the parameter $\beta$ determines infection rate (composed of number of contacts per person per time and the contagion infectivity), while the parameter $\gamma$ determines the recovery rate.

The CC-IB model is derived from this model by assuming a population of $N$ individuals, identified by index ($i = 1, 2, 3, ..., N$). The individuals can switch between **S** and **I** according to probabilistic rules. Each individual can have an individual value for the infection rate $\beta$, the value of the $i$–th individual denoted $\beta_i$. Since the infectivity of a virus is an inherent, non-changing property of a specific virus, modulations of $\beta_i$ directly represent changes in the contact rate in the model. Since infectivity is modeled as constant across individuals, variations in the parameter $\beta_i$ were interpreted solely as differences in contact rates. The model operates on a discrete time-scale. Each timestep, an individual in **S** acquires an infection (and hence switches to **I**) with probability $\beta_i I/N$:

A diseased individual currently residing in **I** recovers (and hence switches to **S**) with a probability that is given by a transition function $f(\theta_i)$ per timestep, where $\theta_i$ denotes the time since the last recovery of the $i$-th individual. $f(\theta_i)$ is defined as the sum of a term representing the ability of the untrained immune system to clear an infection ($c$) and a term representing immunity due to previous exposure to the virus. The term representing specific immunity decays exponentially with rate $d\theta_i$, where $d$ defines the timescale of the immunological memory:

$$f(\theta_i) = a \exp(-d\theta_i) + c$$

Hence, in summary for each individual identified by index $i$, the probabilities to switch from one state to the other within the timestep read as follows:

$$p_{\mathbf{S}\to\mathbf{I}} = \beta_i/N$$
$$p_{\mathbf{I}\to\mathbf{S}} = f(\theta_i)$$

$$= a \exp(-d\theta_i) + c$$

In the simulations, three different parameterizations of $f(\theta_i)$ are considered that correspond to three scenarios "No specific immunity", "Medium specific immunity", "Strong specific immunity" (Fig 2).

The exact values of the parameters a, d, and c in the three scenarios were chosen heuristically in order to produce qualitatively distinct and interpretable regimes of immune response behavior, in particular different recovery dynamics.

The CC-IB model is simulated 10,000 timesteps via Monte Carlo simulations. The population is comprised of 50 "test individuals", that are representative of the population as a whole since they cover the entire parameter range and 950 additional individuals ensuring a sufficient population size. The trajectories of the test individuals are followed on a per-individual-basis in graphical representations. In the test individuals, $\beta_i$ ranges from 0.001 to 0.5 equally spaced on a log scale. In the additional individuals, $\beta_i$ is sampled from a log-normal distribution with $\mu = -4$ and $\sigma = 1$.

Initially, without loss of generality, 10 randomly chosen individuals are infected ($I(0) = 10$), the other individuals are susceptible ($S(0) = 990$). It is assumed that none of the individuals have been exposed to the virus before ($\theta_i = \infty$ for all individuals initially).

A graphical overview of the two presented models and their underlying variants can be found in Fig 3.

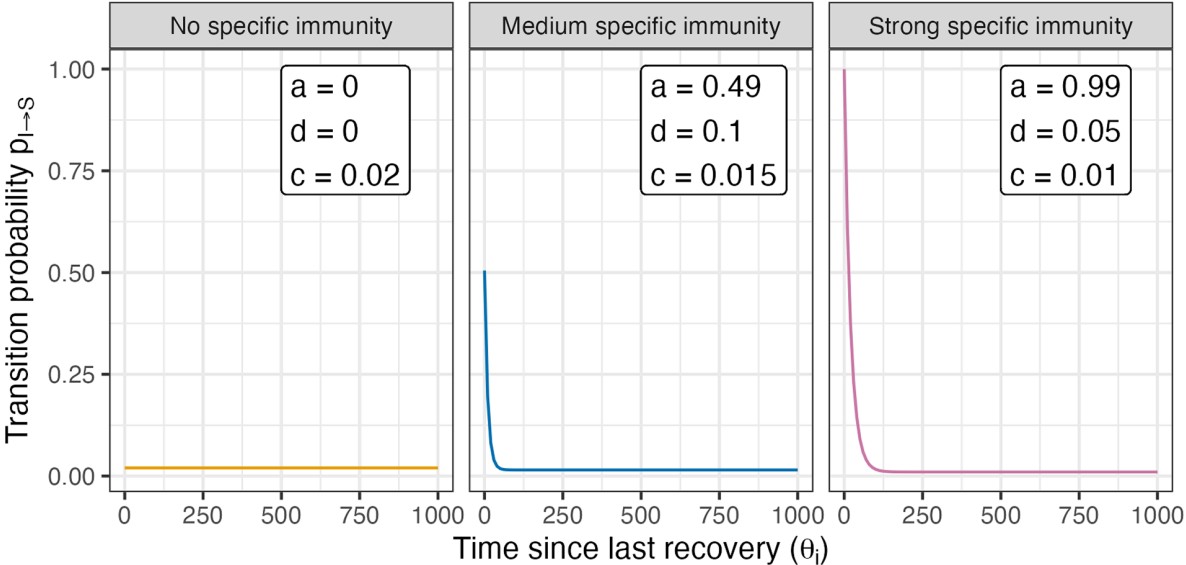

**Fig 2. Transition function for the considered scenarios in the CC-IB model.** The transition function describes the recovery probability $p_{I \to S} = a \exp(-d\theta_i) + c$ (i. e. the probability to switch from the diseased state **I** to the susceptible state **S**). In the first scenario ("No specific immunity"), no training of the immune system is assumed. In the second scenario ("Medium specific affinity"), it is assumed that an infection results in a mild immunity, which is decaying with time. In the third scenario ("Strong specific immunity"), it is assumed that infection results in a strong immunity, which, however, is also decaying with time.

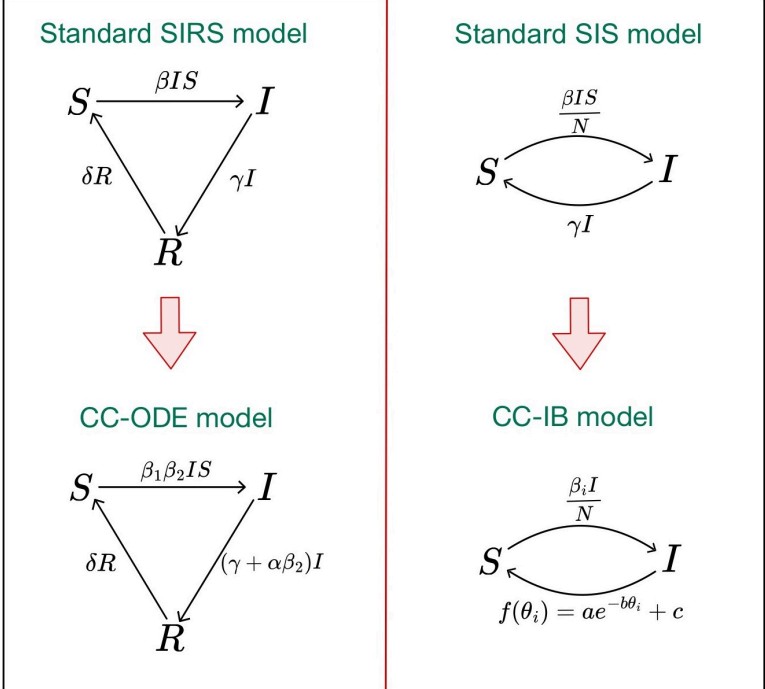

**Fig 3. Model overview.** In this figure, the underlying standard SIRS model (upper left corner) and standard SIS model (upper right corner), as well as the novel developed CC-ODE model (lower left corner) and CC-IB model (lower right corner) are shown. The complete model equations can be found in equation 1 (SIRS model), equation 2 (CC-ODE model), equation 3 (SIS model) and equation 4 (CC-IB model).

# 3 Results

## 3.1 CC-ODE model

The model dynamics for some parameter choices of the immunogenicity $\alpha$ and the contact rate $\beta_2$ are shown in Fig 4. The parameters $\alpha$ and $\beta_2$ are of primary interest in this study and are varied systematically in the following. The other parameters are held constant with $\beta_1 = 0.7$, $\gamma = 0.1$ and $\delta = 0.1$ in order to reduce complexity, but could in principle be estimated from biological data, in case it is available for specific viruses.

As one would expect, with an increasing value of immunogenicity $\alpha$ and constant values for all other parameters, there are more susceptible and less infected individuals. The dynamics for varying contact rate $\beta_2$ appear to be a bit more complex and depend essentially on the choice of $\alpha$. To further investigate this, we have a closer look on the steady states of the ODE system. There are two steady states of the ODE system. One of them is the trivial steady state $(S^*, I^*, R^*) = (1, 0, 0)$, in which there are solely susceptible individuals in **S**, while there are neither infected nor immune individuals in the population. Hence, there is no infection at all that could be spread and the entire population remains susceptible and healthy. The condition

$$\beta_2(\beta_1 - \alpha) - \gamma > 0$$

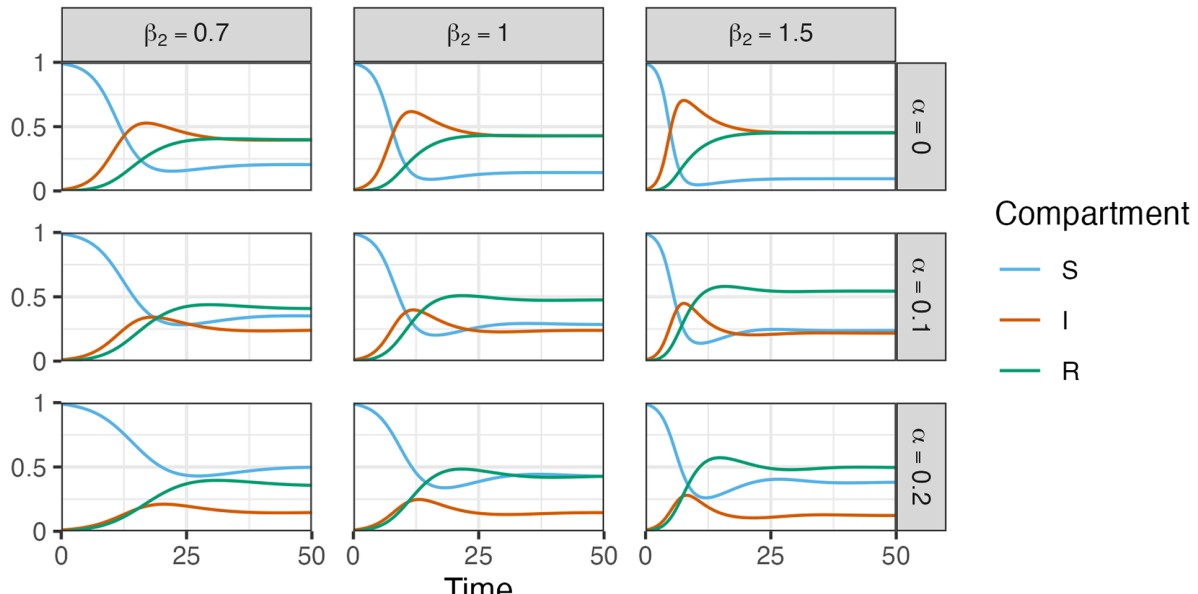

**Fig 4. Solutions of CC-ODE model.** The solutions are depicted for $t \in [0, 50]$ with initial conditions $S(0) = 0.99$, $I(0) = 0.01$, $R(0) = 0$, parameter values $\beta_1 = 0.7$, $\gamma = \delta = 0.1$ and different choices for immunogenicity $\alpha \in \{0, 0.1, 0.2\}$ and contact rate $\beta_2 \in \{0.7, 1.0, 1.5\}$.

ensures instability of the trivial steady state $(S^*, I^*, R^*) = (1, 0, 0)$ and hence the basic reproduction number reads

$$\mathcal{R}_0 = \frac{\beta_1 \beta_2}{\gamma + \alpha \beta_2}.$$

As a consequence, there is a disease outbreak, if $\mathcal{R}_0 > 1$. Otherwise, an introduction of the virus to a small part of the population leads to the extinction of the disease. On the one hand, the higher the probability of infection upon exposure $\beta_1$ and contact rate $\beta_2$, the higher the chance that the disease is breaking out. On the other hand, the greater the immunogenicity $\alpha$ of the virus and the recovery rate $\gamma$, the lower the probability that the disease is breaking out. This coincides well with the intuitive notion of these model parameters.

Additionally, there is another, non-trivial steady state, representing the endemic equilibrium:

$$\begin{pmatrix} S^* \\ I^* \\ R^* \end{pmatrix} = \begin{pmatrix} \dfrac{\gamma + \alpha \beta_2}{\beta_1 \beta_2} \\ \dfrac{\delta(\beta_1 \beta_2 - (\gamma + \alpha \beta_2))}{\beta_1 \beta_2 (\gamma + \alpha \beta_2 + \delta)} \\ \dfrac{(\gamma + \alpha \beta_2)(\beta_1 \beta_2 - (\gamma + \alpha \beta_2))}{\beta_1 \beta_2 (\gamma + \alpha \beta_2 + \delta)} \end{pmatrix} = \begin{pmatrix} \dfrac{1}{\mathcal{R}_0} \\ \dfrac{\delta(\mathcal{R}_0 - 1)}{\mathcal{R}_0((\alpha \beta_2 + \gamma) + \delta)} \\ \dfrac{(\gamma + \alpha \beta_2)(\mathcal{R}_0 - 1)}{\mathcal{R}_0(\gamma + \alpha \beta_2 + \delta)} \end{pmatrix}$$

This expression closely resembles the endemic equilibrium of the classical SIRS model with waning of immunity (see Sect 2.1). The above result follows directly from the classical solution by substituting $\gamma \to \gamma + \alpha \beta_2$ and $\beta \to \beta_1 \beta_2$.

In the following, we focus on this second steady state representing the endemic equilibrium. Thereby, we further investigate $I^*$ and its behavior depending on the parameters $\alpha$ and

$\beta_2$, see Fig 5 a). $I^*$ can be interpreted as the proportion of infected individuals in the long term and is therefore a suitable representation of the mean disease burden after an initial period. For $\alpha > 0$, we obtain curves with one single maximum at

$$\beta_2^* = \frac{\alpha\gamma + \sqrt{\alpha^2\gamma\delta + \alpha\beta_1\gamma^2 + \alpha\beta_1\gamma\delta}}{\alpha\beta_1 - \alpha^2}.$$

Hence, for a contact rate $\beta_2 > \beta_2^*$, there is a decrease of $I^*$ and thus of the number of infections in the long term. Depending on the specific characteristics of a virus and the resulting different parameters in the ODE model, a higher contact rate of the individuals can lead to an overall lower mean disease burden. That is, having overall more contacts helps to reduce the mean disease burden, if there is sufficient development of specific immunity. Interestingly, with increasing probability of infection upon exposure $\beta_1$ the location of the maximum $\beta_2^*$ is decreasing, see Fig 5 b). This means that for pathogens with higher probability of infection upon exposure, it might be the better strategy to have also a higher contact rate to keep the mean disease burden lower. Note, however, that this may not be reasonable for all diseases. The stability analysis of the steady states can be found in the supplementary material S1 File.

## 3.2 CC-IB model

The CC-IB model focuses on the dynamics at the individual level. For the CC-IB model, we evaluate three scenarios corresponding to different patterns of immunity development (no specific immunity, medium specific immunity, strong specific immunity). The number of persons in the compartments **S** and **I** over time is depicted in Fig 6. It can be seen that in all three scenarios, the fraction of infected individuals ($I/N$) oscillates around an equilibrium point.

In order to illustrate the model dynamics, relation between the infection rate and the mean residence time in the two compartments is visualized in Fig 7. For compartment **S**, we can observe that individuals with large values for $\beta_i$ tend to have shorter residence times. This is to be expected, as exposure to the infectious virus is less likely for small values of $\beta_i$. The transition from **S** to **I** is identical for all three scenarios, so that there are no relevant differences

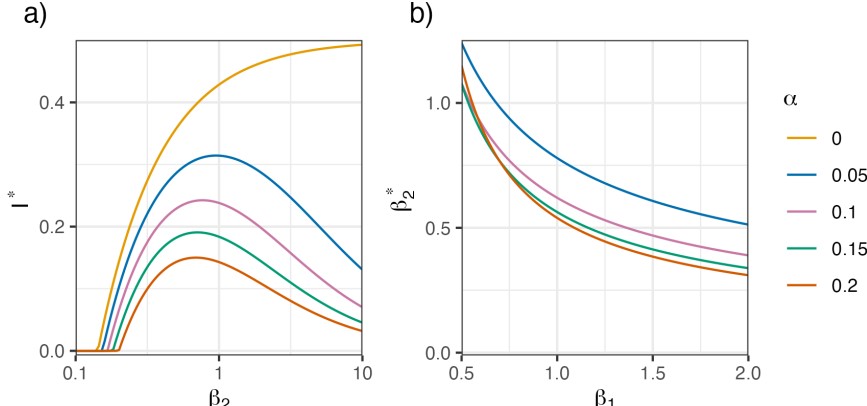

**Fig 5. Analytical results of CC-ODE model.** a) Analytical solutions of steady state component $I^*$ (infected individuals) vs. log-scaled contact rate $\beta_2$ and $\beta_1 = 0.7$. b) Maximum location of steady state $\beta_2^*$ vs. pathogen infectivity $\beta_1$. The colors code for different choices of the immunogenicity $\alpha$. Other parameter values are $\gamma = \delta = 0.1$.

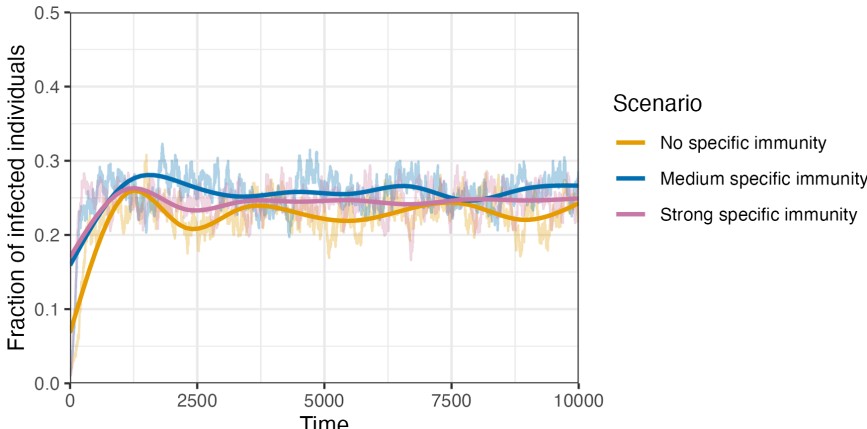

**Fig 6. Dynamics in the compartments in the CC-IB model.** The system oscillates around an infection level of about 25% in all three scenarios. The solid line is based on locally estimated scatter plot smoothing. The qualitative system dynamics are independent of the starting conditions, provided extinction of the disease does not occur (simulations not shown).

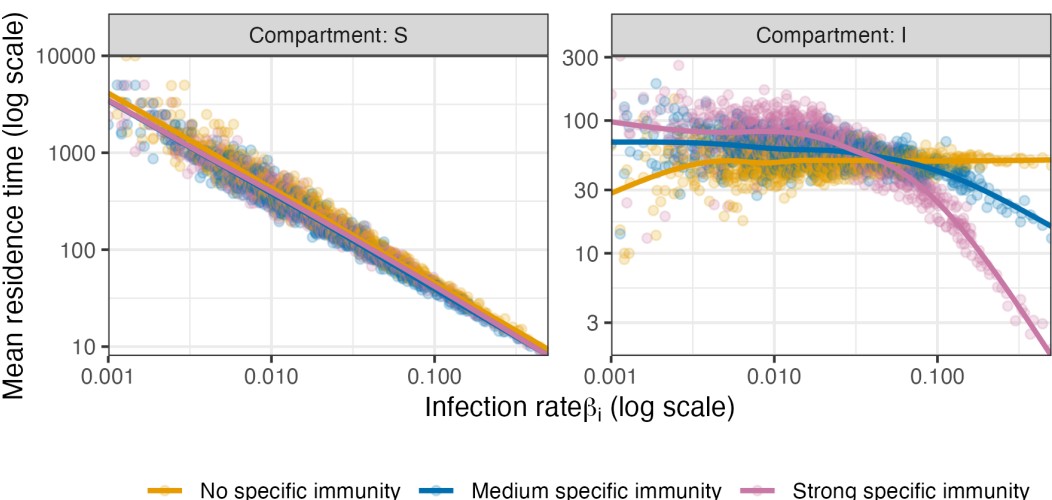

**Fig 7. Mean residence times in the compartments S and I.** For all individuals, the mean residence times in **S** and **I** have been calculated and plotted as individual dots on log scale versus the infection rate $\beta_i$ on log scale. The solid lines are based on locally estimated scatter plot smoothing.

between the three scenarios in this regard. In contrast, the mean residence time in **I** displays important differences between the three scenarios. In scenario 'No specific immunity', the duration of infections (= residence time in **I**) shows no dependence of the infection rate. For scenario "Medium specific immunity", it can be seen that individuals with a higher infection rate (large $\beta_i$) tend to have shorter infections. This relation is even more pronounced in the scenario "Strong specific immunity".

In Fig 8, we show individual trajectories for the time interval from $t = 9000$ to $t = 10000$. It becomes clear that in all three scenarios, individuals with a small $\beta_i$ are rarely infected and individuals with larger $\beta_i$ have infections more frequently. In the first scenario ("No specific immunity"), the duration of infections is independent of $\beta_i$. In the second scenario ("Medium

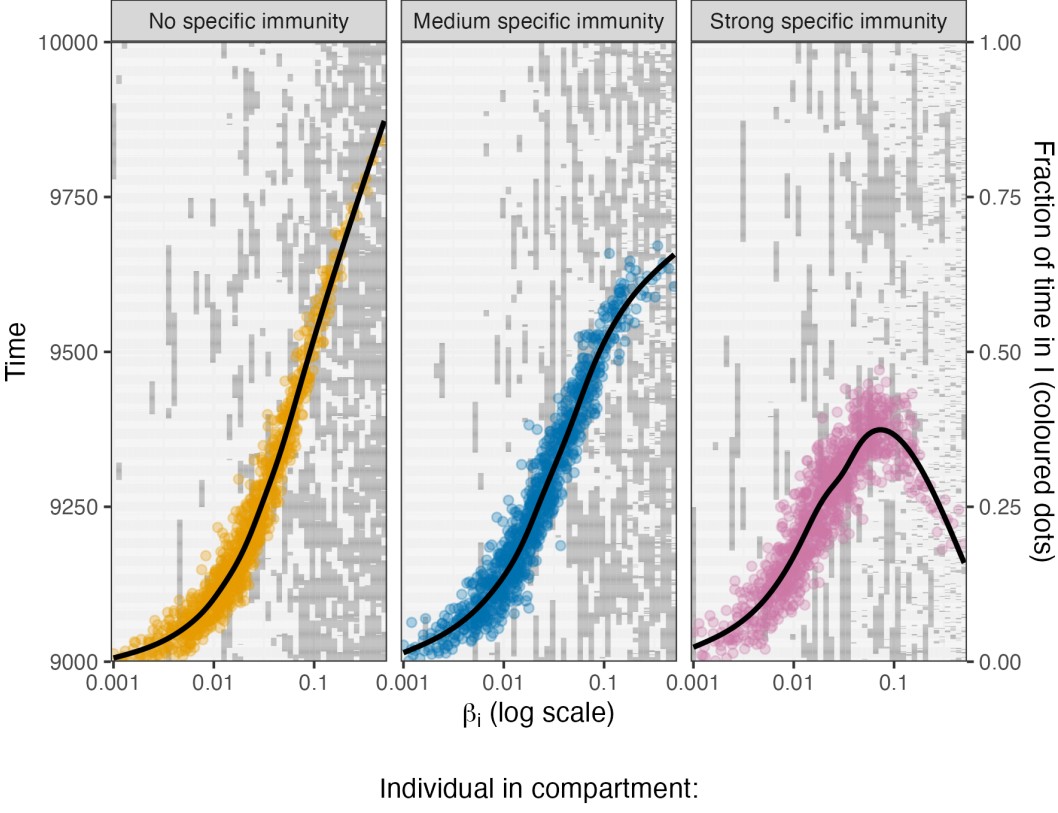

**Fig 8. Trajectories of test individuals and mean disease burden.** In this figure, time is plotted from bottom to top. Each column corresponds to a simulated test-individual with a particular value for the infection rate $\beta_i$. At the times marked dark gray, the individual is in **I**, at the times marked light gray in **S**. The colored dots and the black trend lines indicate the fraction of time spent in **I** with respect to the entire simulation time. The corresponding scale is on the right hand side of the figure.

specific immunity"), infections appear to be shorter for larger values of $\beta_i$. In the third scenario ("Strong specific immunity"), this notion becomes more evident. In this scenario, the total time spent in state **I** first increases with increasing infection rate $\beta_i$. For large values of $\beta_i$, infections are frequent, however, tend to be short in duration.

The fraction of time spent in **I** in the simulation interval ($t$ from 0 to 10000) is chosen as a proxy for mean disease burden. The fraction of time spent in **I** is the product of the number of infections and the mean residence time divided by the total time. In the scenarios "No specific immunity" and "Medium specific immunity", there is a monotonous trend to increase with greater $\beta_i$ (see trend line and colored dots in Fig 8). However, in the scenario "Strong specific immunity", the mean disease burden tends to decline once the infection rate $\beta_i$ (and hence the contact rate) surpasses a certain value of $\beta_i$. Hence, we confirm in our model analysis that also in populations that are heterogeneous with respect to their contact rates, an increased exposure frequency can lead to a decreased mean disease burden for appropriate parameter configurations, as already shown for the CC-ODE model.

## 4 Discussion

The common cold causes considerable human distress and imposes substantial economic costs. It is widely assumed that adopting contact reduction, physical distancing and disinfection measures can reduce infections and that these measures are suited to reduce associated human suffering and economic burden. Although this may be true in many circumstances, it is not clear if this reasoning is universally valid.

To our knowledge, no prior studies have explicitly examined the relationship between exposure frequency and net disease burden. While the underlying idea may appear intuitive, it seems to have received limited attention in the existing literature.

In this paper, we present novel mathematical approaches that study the relation between pathogen exposure frequency and mean disease burden. We deliberately kept the presented approaches simplistic, yet they yield informative insights. Specifically, in this work we introduce an ordinary differential equation model derived from the SIRS model and an individual-based model derived from the SIS model. We apply the models to different theoretical scenarios in order to investigate this relationship.

In both modeling approaches, we demonstrate that for appropriate parameter constellations (i. e. properties of the virus and the population) an increased exposure may lead to a reduced mean disease burden. This can be explained by an efficient training of the immune system in the case of frequent infections. On the contrary, by reducing the number of infections, development of adequate immunity on an individual and population level may be hampered. Therefore, our models predict a reduced mean symptom burden with an increased exposure to a particular common cold virus, both on population and individual level for parameter configurations with high habituation rates. It is important to note that our models rely on the assumption that every exposure to the virus, even if it is not linked to a symptomatic infection (CC-ODE model) or associated with only a very short infection (CC-IB model) leads to full immunization.

For the CC-ODE model, the resulting properties and overall model behavior can be calculated analytically for given parameter configurations as demonstrated in Sect 3.1.

In the CC-ODE model, immune system boosting is assumed to depend in a direct way solely on the contact rate ($\beta_2$), rather than on the probability of infection per contact ($\beta_1$). However, by leading to more frequent infections, an increased $\beta_1$ can indirectly enhance the immunity level, which in turn may increase the recovery rate. This notion reflects a simplifying assumption that repeated exposure—even without leading to full infection and substantial viral replication—can stimulate the immune system. While this assumption allows us to isolate the role of exposure frequency in shaping immune dynamics, we acknowledge that it is a strong assumption. Immune boosting is likely influenced not only by the frequency of contacts but also by the probability and intensity of actual infections, which would be captured by $\beta_1$. We chose this simplified formulation to keep the model as simple as possible and maintain the focus on the impact of contact patterns rather than properties of the virus. Nevertheless, future work could refine this framework by incorporating $\beta_1$-dependent boosting mechanisms, allowing for more realistic, dose- or infection-based immune dynamics.

While an absolute isolation of contagious individuals always leads to zero infections, this radical strategy is connected to great economic cost and personal restrictions. For rather harmless diseases such as common colds, this strategy is certainly not practicable. On the contrary, the better option could be to increase the contact rate in order to reduce the mean disease burden (if practically feasible). We can observe a decreasing mean disease burden with a larger overall contact rate after a certain threshold for most parameter configurations. The optimal strategy highly depends on this threshold and is probably different for each pathogen

and considered population. For most people, the number of contacts of each individual is limited by external circumstances in practice such as occupation and lifestyle and can only be manipulated at a certain cost to the individual. Examining factors that influence contact patterns, several studies have studied the impact of media coverage, such as in the cases of HIV [26] and COVID-19 [27]. These findings suggest that targeted media coverage can influence exposure rates, potentially affecting the spread of a disease within a population. It should be noted here that measures to deliberately increase contact rates are only ethically justifiable if the death rate is close to zero. In the case of common cold pathogens this should usually hold true. However, we definitely do not want to recommend increasing the contact rate for serious illnesses, even if this could lead to fewer infections.

The CC-IB model allows for studying the effects of heterogeneity, e. g. regarding contact behavior and / or development of immunity after exposure. In the current approach presented in Sect 3.2, we introduce heterogeneity exclusively with respect to the contact rate. For reasons of simplicity, the development of immunity and the resulting recovery dynamics are assumed to be identical for all individuals. Furthermore, the individual-based approach adds stochastic aspects, both for infection and recovery. Analogous to the CC-ODE model, the results hint at the possibility that there might be constellations in which an increased exposure leads to a reduced mean disease burden with respect to specific viruses. The CC-ODE model is more targeted at global decision-making issues, that might arise in public health politics. This approach is more suited to derive public health strategies to reduce the mean disease burden. On the contrary, if the concern is to give specific advice to patients, the CC-IB model can give more insightful recommendations. Furthermore, the individual-based model allows to study the dynamics of populations with heterogeneous contact rates, which is not possible with the ordinary differential equation approach presented in this paper. Both models are rather simplistic in design and do not implement all possible facets of interaction between individuals and common cold viruses.

In the CC-IB model, the notion of immunity is captured through a continuously varying, individual-specific state variable rather than a binary classification (susceptible vs. resistant). Individuals with high immunity levels are effectively close to resistant to infection, but they still reside within the broader "susceptible" category in terms of model structure. Hence, our model can be interpreted as a generalization of the SIRS framework, where immune protection is treated as a gradual and dynamically evolving trait.

Even though factors such as viral load and disease severity presumably influence an individual's infectivity, we made the simplifying assumption in our modeling approach that infectivity is constant across infected individuals. As such, variations in the parameter $\beta_i$ were interpreted solely as differences in contact rates. We acknowledge that this is a simplification. Incorporating individual variation in infectivity would certainly be a valuable extension of the model. However, doing so would require the introduction of an additional parameter or distribution, which we chose to avoid at this stage in order to maintain the conceptual clarity and tractability of the model.

The parameters of the two models are related in a qualitative way. The infection rate $\beta$ is described by infectivity $\beta_1$ and the contact rate $\beta_2$ in the CC-ODE model and by $\beta_i$ in the CC-IB model. The specific immunogenicity is given by $\alpha$ and $a$, respectively. The parameter $\gamma$ in the CC-ODE model and the parameter $c$ in the CC-IB model correspond to the unspecific immunity. The time scale / duration of the immunity is modeled by the rate $\delta$ in the population-based approach and by $d$ in the individual-based approach.

In our modeling approach, we prioritized simplicity, following the principle of Occam's razor. We aimed to keep the models as straightforward as possible while maintaining the flexibility to study the relationship between mean disease burden and exposure frequency. Extensions, such as incorporating seasonal forcing and spatial aspects, can be added as needed to fit the model to actual data.

It is important to note that the time scale of re-exposure in our model is not rigidly defined. Rather, we consider a broad range—from days to months or even years—while deliberately excluding dynamics that unfold over an entire human lifespan. This modeling choice reflects our focus on medium-term immune dynamics and justifies the decision to omit host demography (e.g., birth and death processes), which would be more relevant in long-term epidemiological models.

Furthermore, the framework is intentionally pathogen-agnostic. The model is designed to capture general mechanisms of exposure-driven immunity and disease dynamics of the common cold, rather than the specific behavior of any single virus. When sufficient quantitative data on transmission, immunity, and symptom profiles of individual common cold viruses become available, model parameters can be calibrated accordingly, allowing for testable, pathogen-specific predictions. Although the model is inspired by the clinical context of common cold infections, its structure is broadly applicable to other recurrent, self-limiting infections such as viral gastroenteritis.

The primary aim of this study is to investigate if an increased exposure frequency can lead to a reduced mean disease burden in a mathematical modeling framework. At this stage, this is a hypothesis generated by a mathematical modeling approach and at the moment it is not possible to determine how frequently or under what real-world conditions this phenomenon might occur. This would require currently lacking robust empirical data to calibrate the model parameters. Instead, our goal was to propose and discuss a potentially relevant mechanism that consistently explains empirical observations and which merits further quantitative studies.

To assess net disease burden in this generalized setting, we use the fraction of time individuals spend in the infectious state (**I**) as a proxy. While a more comprehensive burden measure—such as one incorporating symptom intensity or severity—would be conceptually desirable, such additions would require more complex modeling and data, which are currently lacking. Future extensions of the framework could certainly include such refinements, especially if longitudinal clinical datasets become available.

In this work, we adopt an aggregated, phenomenological perspective, focusing on the collective dynamics of the common cold syndrome. This simplification is motivated by both the overlapping clinical presentation and the practical limitations in data availability. The model could in principle be extended to describe also the immunological interactions between different viruses, in particular potential cross-immunity between different pathogens.

In the case of common cold pathogens that do not lead to protective immunity (e. g. parainfluenza viruses [28], metapneumoviruses [29]), there is insufficient evidence and data to answer the following hypothetical question: Would a person exposed to a specific common cold pathogen (e. g. a particular rhinovirus) at high frequency (e. g. once a day / hourly) experience persistent corresponding symptoms? Furthermore, the impact of cross-reactivity between different virus serotypes is also not well-studied. For non-common-cold infectious agents, there is a small number of studies systematically investigating intensity of symptoms upon repeated exposure [30–33].

In order to test the results derived in this analysis, different experimental strategies are conceivable. First, in an animal study, animals could be repeatedly infected with a typical common cold virus (e. g. a rhinovirus) in a controlled setting, systematically varying

the time interval between subsequent infections and recording the intensity of the disease symptoms. Appropriate animal models have been described in the literature [34]. Second, humans could be repeatedly infected with different time intervals with a mild common cold virus. In an investigation studying the relation between sleep duration and intensity of symptoms of a common cold, the authors chose such an approach [35]. From our point of view, however, it is questionable whether such a procedure is ethically justifiable for our research question for both animal and human studies, as even generally mild pathogens can lead to more severe courses of disease in rare cases. A third conceivable approach is to collect observational data from groups with different exposure rates to common cold viruses via questionnaires. An attempt could be made to quantify the exposure and symptom burden of common cold infections in a prospective longitudinal study, ideally including participants with a wide range of different contact behaviors. A fourth approach might be related to a more comprehensive surveillance of common cold pathogens by federal agencies, as is already being done (e. g. weekly reports by the Robert-Koch-Institut in Germany - https://influenza.rki.de/Wochenberichte.aspx).

A general increase of contacts in order to stimulate training of the specific immunological defense seems neither feasible nor desirable. Foremost, there are not only mild pathogens in circulation. Viruses such as SARS-CoV-2 and influenza lead to severe and life-threatening infections. The presence of dangerous infectious agents is prohibitive of calling for a general increase of contacts. Nonetheless, the considerations raised in this article raise the question of whether the repertoire of infectious agents to which the immune system is exposed could be specifically targeted in a novel way. In certain environments such as childcare facilities or schools, for example, constellations are conceivable in which avoiding contact could lead to a worsening of the mean disease burden due to common colds. Several studies have examined how contact patterns and disease dynamics are influenced by media coverage, such as in the cases of HIV [26] and COVID-19 [27]. These findings suggest that targeted media coverage can influence exposure rates, potentially affecting the spread of a disease within a population.

In conclusion, a more detailed characterization of disease dynamics of the common cold seems worthwhile. To this end, more detailed observational and experimental data are required in order to facilitate more specific mathematical modeling approaches. More sophisticated mathematical models fitted to more precise data potentially allow to derive recommendations suited to decrease the disease burden associated with the common cold.

## Supporting information

**S1 File. Stability analysis.** This file contains stability analysis for the fixed points.
(PDF)

**S2 File. Source code.** This file contains the source code needed to reproduce all calculations carried out in the entire manuscript.
(HTML)

## Author contributions

**Conceptualization:** Sebastian Gerdes, Michael Rank, Ingmar Glauche, Ingo Roeder.

**Formal analysis:** Sebastian Gerdes, Michael Rank.

**Investigation:** Sebastian Gerdes, Michael Rank.

**Methodology:** Sebastian Gerdes, Michael Rank.

**Supervision:** Ingmar Glauche, Ingo Roeder.

**Writing – original draft:** Sebastian Gerdes, Michael Rank.

**Writing – review & editing:** Sebastian Gerdes, Michael Rank, Ingmar Glauche, Ingo Roeder.

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
