## [Decision Letter · Decision Letter 0]

23 Jan 2025

PONE-D-24-38387Impact of exposure frequency on disease burden of the common cold - a mathematical modeling perspectivePLOS ONE

Dear Dr. Gerdes,

Thank you for submitting your manuscript to PLOS ONE. After careful consideration, we feel that it has merit but does not fully meet PLOS ONE’s publication criteria as it currently stands. Therefore, we invite you to submit a revised version of the manuscript that addresses the points raised during the review process.

Your manuscript was reviewed by two experts in the field. Both reviewers found many important problems in your submission and produced copious comments. It is essential that you respond to all points in the comments.

We look forward to receiving your revised manuscript.

Kind regards,

Yury E Khudyakov, PhD

Academic Editor

PLOS ONE

Journal Requirements:

3. We noted in your submission details that a portion of your manuscript may have been presented or published elsewhere. “The manuscript has been uploaded to a preprint server:

https://www.medrxiv.org/content/10.1101/2024.04.26.24306416v1” Please clarify whether this [conference proceeding or publication] was peer-reviewed and formally published. If this work was previously peer-reviewed and published, in the cover letter please provide the reason that this work does not constitute dual publication and should be included in the current manuscript.

Reviewers' comments:

Reviewer's Responses to Questions

**Comments to the Author**

1. Is the manuscript technically sound, and do the data support the conclusions?

Reviewer #1: Yes

Reviewer #2: No

2. Has the statistical analysis been performed appropriately and rigorously? 

Reviewer #1: N/A

Reviewer #2: No

3. Have the authors made all data underlying the findings in their manuscript fully available?

Reviewer #1: Yes

Reviewer #2: Yes

4. Is the manuscript presented in an intelligible fashion and written in standard English?

Reviewer #1: Yes

Reviewer #2: Yes

5. Review Comments to the Author

Reviewer #1: General points:

This is a well written paper containing competent analyses of mathematical models of viral dynamics. However it is very difficult to assess its relevance to the very important practical issue that it addresses, because:

(i) it refers to few results of empirical studies, and few other studies of the dynamics of respiratory viruses. There is much more relevant literature on cold viruses than the authors cite, and much relevant clinical and modelling work relating to influenza or COVID.

(ii) it is agnostic about the time scale of the effects so is impossible to tell whether the system is achieving a steady state within a short period (which would mean that the neglect of host demography might be OK), or if the period required is comparable or longer than the human lifetime (in which case it would be imperative to consider human birth and death)

The statement that ‘the common cold is a frequent disease’ (line 3) pre-empts the question of whether it is appropriate to treat it as one ‘disease’ (or ‘illness’, noting the different meanings of these words in epidemiology), or as a collection of different infections, each with its own dynamics. Many modellers would treat the common cold as an ensemble of different cross-reacting pathogens, each with long-lived strain-specific immunity that might be described with SIR dynamics. (I strongly suspect some have done so). The authors should explain why they have not chosen this route.

Minor points

Abstract- the reference to ‘this virus’ at the end of the first paragraph is at variance with the explanation later in the paper that the common cold is caused by many different viruses. The assumption that a single, or small set of models, can capture the dynamics of this ensemble of pathogens needs to be spelled out and justified.

The main conclusion (2nd paragraph of the abstract) about increased exposure possible resulting in lower average disease burden needs to be placed in the context of what other research says about this. Was there nothing relevant in the output of the Common Cold Research Unit? (see

https://www.thelancet.com/journals/lancet/article/PIIS0140-6736(05)66722-0/fulltext)

line 3.

Lines 9 -25. The summary of the existing literature on the common cold is cursory

Line 48. The claim about the decline in immunity to cold viruses should be supported by some kind of citation.

Line 68. The SIS model dates back at least to Muench ‘Catalytic models in epidemiology’ (1959).

Line 70. In the classical SIR-model there is no loss of immunity. I think the authors mean SIRS here. There is some repetition in lines 122-125 of what the reader expects to find here.

Lines 71, 74, 76. There are some broken links here.

Lines 81-82. I would be surprised if it is true that ‘the exposure frequency to particular viruses …. has not been explicitly studied.’ Absence of proof is not proof of absence, but have the authors really looked in the literature for this?

Lines 109 and 115. The internal references are broken.

Line 119. What is ‘Hello ??’

Line 163 ‘depicts’ should be ‘measures’

Line 197. Why should the rate of recovery depend on the time since the last recovery?

Line 202. The explanation that index i identifies each individual should be given with the first occurrence of the index. ‘probabilities to switch …… to the other’ should read . ‘probabilities to switch …… to the other within the timestep’

Line 259 broken link.

Line 288. The fraction of time spent in I is a poor proxy for mean disease burden, since it conflates duration of what might be sub-clinical infections with severity. The authors should at least attempt to justify and explain using this measure.

Line 325 broken link

Line 372 The assertion about protective immunity should be supported by a citation.

Reviewer #2: The manuscript entitled “Impact of exposure frequency on disease burden of the common cold - a mathematical modeling perspective” investigates the impact of exposure frequency on infectious disease burden. The central argument, as I understand it, is that increased exposure boosts immunity via a specific immune response, and that beyond a certain threshold this heightened immunity may outcompete infection rates and ultimately lower disease burden. To study this hypothesis, the authors use two different modeling approaches:

an ODE-based (deterministic) SIRS model that includes a contact-frequency-dependent recovery rate; and

an individual-based (stochastic) SIS model in which the recovery rate wanes over time since an individual’s last infection.

While the idea is interesting, I have concerns that the chosen model configurations may be either inappropriate or too vague to support the authors’ conclusions.

In the ODE model, the transmission rate is first decomposed into a contact rate component (β_2) and an infection probability per contact component (β_1). The recovery rate is then augmented by a new term that is assumed to be boosted by the specific immune response in direct proportion to the contact rate β_2. As stated above, the assumption is that greater contact leads to more infections, which leads to heightened immunity, which leads to more rapid recovery. However, it is not clear to me why β_2 has been specifically isolated to investigate this effect. Would the same argument not equally apply to the transmission rate component β_1, the probability of infection per contact? In general, immunity should be boosted with increased infection - independent of whether infection was boosted as a result of increased contact (β_2) or infection probability (β_1). I understand that the focus of the investigation was the impact of exposure rates, but I am concerned that β_1 and β_2 are not so easily separated, and that the boosted recovery rate should depend on both quantities.

With respect to the individual-based SIS model, the principal issue is the seeming arbitrariness in how it and its parameters are configured. The manuscript mentions “test” subjects with transmission rates distinct from those of the other individuals, yet the motivation for these separate rates remains unclear. I am also unsure about the rationale behind the chosen parameter ranges for the test subjects, the distribution used for the remaining population, and the specific values of a, d, and c parameters across the “no,” “medium,” and “strong” model configurations. Even acknowledging that this model is not tailored to any specific pathogen (though the common cold is suggested as a possible example), the lack of a clear explanation or justification for these parameter choices makes it difficult to interpret the results. The only apparent rationale is to align the three residence time curves shown in the left panel of Figure 7.

In summary, while the premise of the study is intriguing, the current model implementations suffer from ambiguity in their parameter choices and their selective focus on contact rate as the driver of increased immunity. Because these issues cast doubt on the authors’ conclusions, I cannot recommend this paper for publication in its current form. The models, as presented, appear either insufficiently defined or too vague to be of practical use.

I encourage the authors to consider revising both models to better justify the assumptions made and to clarify parameter selection and modelling decisions. Doing so could strengthen the manuscript and make its conclusions more convincing.

In addition to the general comments above, I also have the following minor comments:

The manuscript would benefit from additional proof-reading. There were many unresolved references and some erroneous text (see e.g., l119)

The paragraph on lines 44-50 could probably benefit from additional references – particularly the claim that “an increased number of sick days due to common colds was observed in the communities where the contact rates had been lower in the preceding years”

The sentence on l70: Do you mean SIS or SIRS model? For one, I don’t believe the classical SIR model incorporates loss of immunity. Even in the SIRS model the loss of immunity is not instantaneous, it is exponentially-distributed.

L112: Even if the R compartment is not explicitly modelled, would it not still be an SIRS-like model is there is an immune state? Isn’t infection, I, also an individual-based state?

I think the expression given for R_0 given between lines 235 and 236 is incorrect. First, the expression given suggests that this quantity would be negative if β_1<α, which is unphysical. Second, the model you propose is in effect still an SIRS model with a new recovery rate γ^*=γ+αβ_2 and a decomposed transmission rate β=β_1 β_2. In this case, I think the reproduction number should be (assuming N=1)

R_0=(β_1 β_2)/(γ+αβ_2 )

In the expression for the endemic state there is in an error in bracket term of the numerator for R^*.

L268-269: Do you mean that individuals with large β_i values tend to have lower residence times (i.e., residence time and transmission rate are negatively correlated)?

6. PLOS authors have the option to publish the peer review history of their article (what does this mean?). If published, this will include your full peer review and any attached files.

Reviewer #1: No

Reviewer #2: **Yes: **Michael T Meehan

---

## [Author Response · Author response to Decision Letter 1]

11 Jul 2025

Please note that we have attached additional files "response_to_reviewer_1.pdf" and "response_to_reviewer_2.pdf" with nicer formatting.

Response to reviewer 1

Reviewer 1: This is a well written paper containing competent analyses of mathematical models of viral dynamics. However it is very difficult to assess its relevance to the very important practical issue that it addresses, because:

(i) it refers to few results of empirical studies, and few other studies of the dynamics of respiratory viruses. There is much more relevant literature on cold viruses than the authors cite, and much relevant clinical and modelling work relating to influenza or COVID.

Response: We fully agree that the body of literature on respiratory viruses—particularly concerning rhinoviruses, influenza, and SARS-CoV-2—is extensive and rapidly evolving. In our original manuscript, we intentionally focused on a limited number of key references in order to maintain clarity and focus within the scope of our study. However, we recognize that providing a broader context enhances the robustness of our argument.

In response to the reviewer’s suggestion, we have now expanded the citations in the background section to include additional empirical and modeling studies relevant to the dynamics of respiratory virus transmission, including recent work on influenza and COVID-19. These additions provide a more comprehensive framing of our approach and underscore the relevance of our model in relation to existing literature (lines 100–105 in file “article_diff.pdf”).

Reviewer 1: (ii) it is agnostic about the time scale of the effects so is impossible to tell whether the system is achieving a steady state within a short period (which would mean that the neglect of host demography might be OK), or if the period required is comparable or longer than the human lifetime (in which case it would be imperative to consider human birth and death)

Response: In the present version of the model, we intentionally refrained from assigning a fixed time scale to the dynamics due to the lack of precise empirical data on key parameters such as transmission rates, duration of immunity, and contact patterns. Our primary aim was to explore the qualitative behavior of the system under general assumptions, rather than to make precise quantitative predictions over a defined time horizon. In typical cases, the duration of a symptomatic common cold infection is on the order of one week. The frequency of re-exposure, depending on individual behavior and environmental context, might plausibly range from several days to multiple years.

However, we agree that the time scale has important implications for model interpretation, particularly with regard to the relevance of host demography. To address this, we have added a more detailed discussion in the limitations section (lines 454 – 459 in file “article_diff.pdf”), suggesting how the model could be extended in future work to include demographic dynamics if long-term behavior is to be studied.

Reviewer 1: The statement that ‘the common cold is a frequent disease’ (line 3) pre-empts the question of whether it is appropriate to treat it as one ‘disease’ (or ‘illness’, noting the different meanings of these words in epidemiology), or as a collection of different infections, each with its own dynamics. Many modellers would treat the common cold as an ensemble of different cross-reacting pathogens, each with long-lived strain-specific immunity that might be described with SIR dynamics. (I strongly suspect some have done so). The authors should explain why they have not chosen this route.

Response: In general, we do indeed conceptualize the common cold as a disease that may result from infections with different common cold viruses over time. In our modeling frame-work, we neglect immunological interactions between different pathogens and look at an exemplary virus inducing a “common cold syndrome". The biological properties (infectivity, recovery dynamics) of this virus may be varied within our modeling framework. We deliberately wanted to present a simple modeling approach, at this stage of the project neglecting potential immunological interactions between different viruses, in particular development of cross-immunity. We have stated this notion more explicitly (see lines 96-98 and lines 460 – 465 and lines 472 – 477 in file “article_diff.pdf”).

Reviewer 1: Abstract- the reference to ‘this virus’ at the end of the first paragraph is at variance with the explanation later in the paper that the common cold is caused by many different viruses. The assumption that a single, or small set of models, can capture the dynamics of this ensemble of pathogens needs to be spelled out and justified.

Response: We acknowledge that the phrasing in the abstract may have unintentionally implied that the common cold is caused by a single virus. To avoid this confusion, we have revised the relevant sentence in the abstract (see page 1, end of first paragraph of the abstract) to clarify that the common cold is a clinical syndrome resulting from a diverse set of viral pathogens.

Furthermore, we have added a sentence to explicitly state our modeling assumption: rather than modeling each virus individually, we adopt a simplified framework that captures the aggregate epidemiological behavior of the ensemble of pathogens. This choice is motivated by the overlapping clinical features, shared transmission routes, and challenges in resolving strain-specific dynamics due to limited data. We now also include a brief justification of this approach in the introduction (see lines 135 - 146).

Reviewer 1: The main conclusion (2nd paragraph of the abstract) about increased exposure possible resulting in lower average disease burden needs to be placed in the context of what other research says about this. Was there nothing relevant in the output of the Common Cold Research Unit? (see

https://www.thelancet.com/journals/lancet/article/PIIS0140-6736(05)66722-0/fulltext)

line 3.

Response: We thank the reviewer for this valuable suggestion. We agree that placing our findings in the context of previous empirical work is important. Following the reviewer’s advice, we carefully reviewed the output of the Common Cold Research Unit, including the article referenced. While the work from this unit provides important insights into susceptibility, transmission, and reinfection patterns, we did not find studies that directly examine the possibility that increased exposure frequency might reduce average disease burden at the population level, as suggested by our model.

To clarify this, we have revised the discussion section (see lines 363 - 367) to explicitly acknowledge this gap in the literature, and to highlight our conclusion as a hypothesis-generating result that may warrant further empirical investigation. We also included a reference to the Common Cold Research Unit to better situate our work in the historical context of cold-related research.

Reviewer 1: Lines 9 - 25. The summary of the existing literature on the common cold is cursory

Response: We recognize that our initial summary of the literature was concise. As the body of research on the common cold is indeed extensive—spanning clinical, epidemiological, immunological, and modeling studies—we chose to keep our overview concise to maintain focus on the specific scope of our study – in particular in view of lack of studies that address points very similar to the goals of this work.

Reviewer 1: Line 48. The claim about the decline in immunity to cold viruses should be supported by some kind of citation.

Response: We added citations substantiating our claims (line 63).

Reviewer 1: Line 68. The SIS model dates back at least to Muench ‘Catalytic models in epidemiology’ (1959).

Response: Thanks a lot for the hint. We have adjusted our manuscript accordingly.

Reviewer 1: Line 70. In the classical SIR-model there is no loss of immunity. I think the authors mean SIRS here. There is some repetition in lines 122-125 of what the reader expects to find here.

Response: Thanks for pointing out the incorrectness at our end. We have adjusted our manuscript accordingly.

Reviewer 1: Lines 71, 74, 76. There are some broken links here. Lines 109 and 115. The internal references are broken.

Response: We apologize for the inconvenience and have fixed the broken links.

Reviewer 1: Lines 81-82. I would be surprised if it is true that ‘the exposure frequency to particular viruses …. has not been explicitly studied.’ Absence of proof is not proof of absence, but have the authors really looked in the literature for this?

Response: In preparing this manuscript, we conducted an extensive literature search across both clinical and modeling domains. While several studies touch on related aspects—such as the duration of immunity, viral interference, or cross-reactivity—we did not find studies that systematically investigate the specific hypothesis that higher exposure frequency could result in a lower long-term average disease burden under certain conditions. A possible explanation might be that the point is too trivial – however, we believe it should nonetheless be explicitly included in the scientific discourse.

Reviewer 1: Line 119. What is ‘Hello ??’. Line 163 ‘depicts’ should be ‘measures’.

Response: We have fixed the issues.

Reviewer 1: Line 197. Why should the rate of recovery depend on the time since the last recovery?

Response: In our models, we have incorporated immunity waning effects. Immunity waning is known to not only have an infect on the probability of reinfection but also on the course of the disease (for common colds https://doi.org/10.3389/falgy.2023.1224988, for SARS-CoV2 https://doi.org/10.1038/s41586-024-08511-9).

Reviewer 1: Line 202. The explanation that index i identifies each individual should be given with the first occurrence of the index. ‘probabilities to switch …… to the other’ should read . ‘probabilities to switch …… to the other within the timestep’

Response: We have more explicitly introduced the meaning of the index and added the words “within the timestep” to clarify!

Reviewer 1: Line 259 broken link.

Response: We have fixed the issue.

Reviewer 1: Line 288. The fraction of time spent in I is a poor proxy for mean disease burden, since it conflates duration of what might be sub-clinical infections with severity. The authors should at least attempt to justify and explain using this measure.

Response: We agree that the fraction of time spent in the infectious state (I) does not capture all dimensions of disease burden, particularly not clinical severity or subclinical presentation. However, in the absence of detailed pathogen-specific or host-specific data on symptom severity, we chose this quantity as a parsimonious, population-level proxy for the average burden of disease. This measure reflects the cumulative incidence and duration of infectious episodes, which we believe still captures an important component of the public health impact, especially in the context of asymptomatic or mildly symptomatic infections that nonetheless contribute to transmission and morbidity.

We acknowledge that more nuanced metrics (e.g., incorporating intensity, symptom scores, or weighted health outcomes such as “quality-adjusted life years”) would provide a richer picture, but incorporating these would require a more complex and highly parameterized model, which was beyond the scope of our present analysis. We now explicitly state this limitation and the rationale for our choice in the discussion section (see lines 466 - 471).

Reviewer 1: Line 325 broken link

Response: We have fixed the issue.

Reviewer 1: Line 372 The assertion about protective immunity should be supported by a citation.

Response: We have added supporting citations.

Response to reviewer 2

Reviewer 2: The manuscript entitled “Impact of exposure frequency on disease burden of the common cold - a mathematical modeling perspective” investigates the impact of exposure frequency on infectious disease burden. The central argument, as I understand it, is that increased exposure boosts immunity via a specific immune response, and that beyond a certain threshold this heightened immunity may outcompete infection rates and ultimately lower disease burden. To study this hypothesis, the authors use two different modeling approaches:

an ODE-based (deterministic) SIRS model that includes a contact-frequency-dependent recovery rate; and

an individual-based (stochastic) SIS model in which the recovery rate wanes over time since an individual’s last infection.

While the idea is interesting, I have concerns that the chosen model configurations may be either inappropriate or too vague to support the authors’ conclusions.

In the ODE model, the transmission rate is first decomposed into a contact rate component (β_2) and an infection probability per contact component (β_1). The recovery rate is then augmented by a new term that is assumed to be boosted by the specific immune response in direct proportion to the contact rate β_2. As stated above, the assumption is that greater contact leads to more infections, which leads to heightened immunity, which leads to more rapid recovery. However, it is not clear to me why β_2 has been specifically isolated to investigate this effect. Would the same argument not equally apply to the transmission rate component β_1, the probability of infection per contact? In general, immunity should be boosted with increased infection - independent of whether infection was boosted as a result of increased contact (β_2) or infection probability (β_1). I understand that the focus of the investigation was the impact of exposure rates, but I am concerned that β_1 and β_2 are not so easily separated, and that the boosted recovery rate should depend on both quantities.

Response: We agree that both components of the transmission rate—β₁ (infection probability per contact) and β₂ (contact rate)—contribute to overall infection risk and, by extension, to immune system stimulation.

In our model, we focused on β₂ as the driver of immune boosting because it captures the behavioral or environmental aspect of exposure: individuals who interact more frequently with others are more likely to be repeatedly exposed to pathogens. In contrast, β₁ is typically interpreted as a biological property of the pathogen-host interaction (e.g., susceptibility, virulence, mucosal barrier integrity), which was not varied in our study.

That said, we agree that it would be both reasonable and biologically plausible to include β₁ as an additional or alternative driver of immune boosting—especially in models tailored to specific pathogens or datasets. For instance, the immune-stimulating term (γ + αβ₂)*I could be extended to include a β₁-dependent component, such as (γ + α₁β₁ + α₂β₂)*I, in future versions of the model.

Our current model represents a simplified conceptual structure designed to isolate the effect of varying exposure frequency under minimal assumptions. We chose to vary β₂ in this first step to maintain transparency and interpretability. We now clarify this rationale in the discussion section of the manuscript and highlight the possibility of extending the immune feedback mechanism to include both β₁ and β₂ in future work (see lines 220 – 224 and lines 387 – 399 in file “article_diff.pdf”).

Reviewer 2: With respect to the individual-based SIS model, the principal issue is the seeming arbitrariness in how it and its parameters are configured. The manuscript mentions “test” subjects with transmission rates distinct from those of the other individuals, yet the motivation for these separate rates remains unclear. I am also unsure about the rationale behind the chosen parameter ranges for the test subjects, the distribution used for the remaining population, and the specific values of a, d, and c parameters across the “no,” “medium,” and “strong” model configurations. Even acknowledging that this model is not tailored to any specific pathogen (though the common cold is suggested as a possible example), the lack of a

---

## [Decision Letter · Decision Letter 1]

5 Aug 2025

PONE-D-24-38387R1Impact of exposure frequency on disease burden of the common cold - a mathematical modeling perspectivePLOS ONE

Dear Dr. Gerdes,

Thank you for submitting your manuscript to PLOS ONE. After careful consideration, we feel that it has merit but does not fully meet PLOS ONE’s publication criteria as it currently stands. Therefore, we invite you to submit a revised version of the manuscript that addresses the points raised during the review process.

Your revised manuscript was reviewed by one original reviewer who still identified many important remaining problems in your work. Please carefully review the attached comments and provide point-by-point responses. 

We look forward to receiving your revised manuscript.

Kind regards,

Yury E Khudyakov, PhD

Academic Editor

PLOS ONE

Journal Requirements:

Reviewers' comments:

Reviewer's Responses to Questions

**Comments to the Author**

1. If the authors have adequately addressed your comments raised in a previous round of review and you feel that this manuscript is now acceptable for publication, you may indicate that here to bypass the “Comments to the Author” section, enter your conflict of interest statement in the “Confidential to Editor” section, and submit your "Accept" recommendation.

Reviewer #2: (No Response)

2. Is the manuscript technically sound, and do the data support the conclusions?

Reviewer #2: No

3. Has the statistical analysis been performed appropriately and rigorously? 

Reviewer #2: N/A

4. Have the authors made all data underlying the findings in their manuscript fully available?

Reviewer #2: Yes

5. Is the manuscript presented in an intelligible fashion and written in standard English?

Reviewer #2: Yes

6. Review Comments to the Author

Reviewer #2: I appreciate the authors efforts to provide considered and eloquent responses to each item raised; however, I am concerned that some issues remain. Primarily, it is still difficult to know the extent to which the generated results apply to common cold viruses. The main result of the paper seems to be captured by the statement "we can observe a decreasing mean disease burden with a larger overall contact rate after a certain threshold for most parameter configurations". However, it is unclear what fraction of these "parameter configurations" are realized in practice. The authors acknowledge that "the framework is intentionally pathogen agnostic" and, without informed parameters, I would argue that this characterizes the analyses themselves - despite the title and narrative repeatedly anchoring to common cold pathogens.

Perhaps the authors observation that "Although the common cold can be caused by a wide variety of viruses, the clinical manifestations - particularly the duration and characteristics of symptoms - often show substantial overlap" could provide some opportunity to constrain or contextualize their model parameters, and in turn their results.

In addition to these general comments, I also have some specific comments below:

l87: Could the authors elaborate on the sense in which immunity loss is instantaneous? In the standard SIRS model the immune period is exponentially distributed

l196-197: The authors state "there is no evidence that the infectivity of the pathogen should also affect the recovery rate". This seems to be contradicted by l392-394 of the discussion: "Immune boosting is likely influenced not only by the frequency of contacts but also by the probability and intensity of actual infections"

l242-243: Related to the previous comment, I disagree with the statement that infectivity is an inherent, non-changing property of a specific virus, and that modulations of \beta_i directly represent changes in the contact rate in the model. Is it not possible that individual variation in viral load and severity would yield modulations in \beta_i independent of contact rates?

l278: Full stop missing after "Fig. 4"

l297: Repeated "the"

l300: I still worry there is an error in the endemic solutions. I think that the extra (separate) \gamma term in the numerator for the R* state is unnecessary. The authors (and readers) may find it helpful to re-express these solutions in terms of the reproduction number and compare the results with standard SIRS model solutions.

7. PLOS authors have the option to publish the peer review history of their article (what does this mean?). If published, this will include your full peer review and any attached files.

Reviewer #2: **Yes: **Michael T Meehan

---

## [Author Response · Author response to Decision Letter 2]

14 Sep 2025

Response to reviewer 2

(An additional file, reply_to_reviewer_2.pdf, is provided for better legibility.)

Reviewer #2: I appreciate the authors efforts to provide considered and eloquent responses to each item raised; however, I am concerned that some issues remain. Primarily, it is still difficult to know the extent to which the generated results apply to common cold viruses. The main result of the paper seems to be captured by the statement "we can observe a decreasing mean disease burden with a larger overall contact rate after a certain threshold for most parameter configurations". However, it is unclear what fraction of these "parameter configurations" are realized in practice. The authors acknowledge that "the framework is intentionally pathogen agnostic" and, without informed parameters, I would argue that this characterizes the analyses themselves - despite the title and narrative repeatedly anchoring to common cold pathogens.

Response: We fully acknowledge that the current model, while inspired by the clinical context of common cold infections, is fundamentally theoretical and pathogen-agnostic. More generally, the model could readily be applied to other common infections, such as viral gastroenteritis. Our primary aim was not to assert that a threshold-dependent reduction in mean disease burden occurs in most parameter configurations, but rather to demonstrate that such an effect is possible within the model—i.e., it emerges under certain plausible configurations of parameters.

At this stage, we are not in a position to determine how frequently or under what real-world conditions this phenomenon might occur, as this would require currently lacking robust empirical data to thoroughly calibrate all the model parameters. Instead, our goal was to propose and discuss a potentially relevant mechanism that consistently explains empirical observations and which merits further quantitative studies. We hope this modeling insight can serve as a foundation for future studies aimed at quantifying this effect in specific epidemiological contexts.

We have added an additional paragraph discussing these aspects in the discussion section (ll475-485).

Reviewer #2: Perhaps the authors observation that "Although the common cold can be caused by a wide variety of viruses, the clinical manifestations - particularly the duration and characteristics of symptoms - often show substantial overlap" could provide some opportunity to constrain or contextualize their model parameters, and in turn their results.

Response: We appreciate the reviewer’s suggestion and agree that the observed overlap in clinical manifestations of common cold pathogens could, in principle, help to constrain or contextualize model parameters. However, given the current level of generality in our model and the lack of robust, pathogen-agnostic epidemiological data on relevant parameters (such as reinfection rates, immunity duration, or exposure frequencies) we believe that applying such constraints at this stage would be premature. Our goal in this work was to explore general, qualitative dynamics rather than to produce calibrated, pathogen-specific predictions (ll475-485). We see parameter refinement based on empirical data as an important step for future, more targeted model applications.

Reviewer # 2: l87: Could the authors elaborate on the sense in which immunity loss is instantaneous? In the standard SIRS model the immune period is exponentially distributed

Response: We apologize for the lack of clarity in our original phrasing. In the standard SIRS model, as correctly noted, the duration of immunity follows an exponential distribution, and the transition from the immune (R) to the susceptible (S) compartment occurs as a stochastic event. What we intended to highlight was that immunity in such models is typically represented as a binary state, in which individuals are either fully immune or fully susceptible, with an abrupt transition between these states once immunity "expires."

In contrast, our intention was to model immunity as a gradual, continuously varying property that wanes over time rather than being lost instantaneously when the cells transit from one to another state. This continuous representation allows for intermediate levels of immunity and more nuanced dynamics, which may better reflect real-world immunological processes, particularly in the context of repeated exposures to common pathogens. We have revised the manuscript to clarify this point (ll75-78).

Reviewer # 2: l196-197: The authors state "there is no evidence that the infectivity of the pathogen should also affect the recovery rate". This seems to be contradicted by l392-394 of the discussion: "Immune boosting is likely influenced not only by the frequency of contacts but also by the probability and intensity of actual infections"

Response: We agree that there is a tension between these two statements and appreciate the reviewer’s careful reading. We have revised the wording to express our reasoning more clearly and consistently (l192, ll388-392). Specifically, in the context of our model, a higher infectivity (β₁) does not directly affect the recovery rate. However, by leading to more frequent infections, it can indirectly enhance the immunity level, which in turn may increase the recovery rate. We hope this clarification resolves the ambiguity.

Reviewer # 2: l242-243: Related to the previous comment, I disagree with the statement that infectivity is an inherent, non-changing property of a specific virus, and that modulations of \beta_i directly represent changes in the contact rate in the model. Is it not possible that individual variation in viral load and severity would yield modulations in \beta_i independent of contact rates?

Response: We agree that factors such as viral load and disease severity can influence an individual’s infectivity. In our modeling approach, we made the simplifying assumption that infectivity is constant across infected individuals. As such, variations in the parameter $\beta_i$ were interpreted solely as differences in contact rates.

We acknowledge that this is a simplification. Incorporating individual variation in infectivity would certainly be a valuable extension of the model. However, doing so would require the introduction of an additional parameter or distribution, which we chose to avoid at this stage in order to maintain the conceptual clarity and tractability of the model. We have now clarified this modeling assumption in the main text (see ll239-241) and added a note to the discussion section (see ll445-452).

Reviewer # 2: l278: Full stop missing after "Fig. 4"

Response: We have added the full stop.

Reviewer # 2: l297: Repeated "the"

Response: We have deleted the extra “the”.

Reviewer # 2: l300: I still worry there is an error in the endemic solutions. I think that the extra (separate) \gamma term in the numerator for the R* state is unnecessary. The authors (and readers) may find it helpful to re-express these solutions in terms of the reproduction number and compare the results with standard SIRS model solutions.

Response: We have now corrected the expression for the endemic equilibrium and apologize for having overlooked this issue in the first round of revisions. In addition, we have included the endemic solution for the classical SIRS model (ll184-186) with waning immunity for comparison. Since our ODE formulation is based on a straightforward parameter substitution (γ → γ + αβ₂ and β → β₁β₂), the structure of the analytical results is preserved. The classical SIRS model emerges as a special case of our framework when αβ₂ = 0 and β is replaced by any chosen value of β₁β₂ (ll298-304).

---

## [Decision Letter · Decision Letter 2]

30 Sep 2025

Impact of exposure frequency on disease burden of the common cold - a mathematical modeling perspective

PONE-D-24-38387R2

Dear Dr. Gerdes,

We’re pleased to inform you that your manuscript has been judged scientifically suitable for publication and will be formally accepted for publication once it meets all outstanding technical requirements.

Kind regards,

Yury E Khudyakov, PhD

Academic Editor

PLOS ONE

Additional Editor Comments (optional):

Reviewers' comments:

Reviewer's Responses to Questions

**Comments to the Author**

1. If the authors have adequately addressed your comments raised in a previous round of review and you feel that this manuscript is now acceptable for publication, you may indicate that here to bypass the “Comments to the Author” section, enter your conflict of interest statement in the “Confidential to Editor” section, and submit your "Accept" recommendation.

Reviewer #2: (No Response)

2. Is the manuscript technically sound, and do the data support the conclusions?

Reviewer #2: Yes

3. Has the statistical analysis been performed appropriately and rigorously? 

Reviewer #2: Yes

4. Have the authors made all data underlying the findings in their manuscript fully available?

Reviewer #2: Yes

5. Is the manuscript presented in an intelligible fashion and written in standard English?

Reviewer #2: Yes

6. Review Comments to the Author

Reviewer #2: I am satisfied with the latest version of the manuscript and appreciate the authors’ efforts addressing the latest round of comments.

7. PLOS authors have the option to publish the peer review history of their article (what does this mean?). If published, this will include your full peer review and any attached files.

Reviewer #2: **Yes: **Michael T. Meehan

---

## [Editor Report · Acceptance letter]

PONE-D-24-38387R2

PLOS ONE

Dear Dr. Gerdes,

I'm pleased to inform you that your manuscript has been deemed suitable for publication in PLOS ONE. Congratulations! Your manuscript is now being handed over to our production team.

Kind regards,

on behalf of

Dr. Yury E Khudyakov

Academic Editor

PLOS ONE